# Dietary Patterns and Risk of Obesity and Early Childhood Caries in Australian Toddlers: Findings from an Australian Cohort Study

**DOI:** 10.3390/nu11112828

**Published:** 2019-11-19

**Authors:** Lucinda K. Bell, Celeste Schammer, Gemma Devenish, Diep Ha, Murray W. Thomson, John A. Spencer, Loc G. Do, Jane A. Scott, Rebecca K. Golley

**Affiliations:** 1Caring Futures Institute, College of Nursing and Health Sciences, Flinders University, Adelaide 5001, Australia; lucy.bell@flinders.edu.au (L.K.B.); celesteschammer@gmail.com (C.S.); 2School of Public Health, Curtin University, Perth 6102, Australia; gemma.devenish@curtin.edu.au (G.D.);; 3Australian Research Centre for Population Oral Health, School of Dentistry, The University of Adelaide, Adelaide 5000, Australia; diep.ha@adelaide.edu.au (D.H.); john.spencer@adelaide.edu.au (J.A.S.); loc.do@adelaide.edu.au (L.G.D.); 4Division of Health Sciences, University of Otago, Dunedin 9016, New Zealand; murray.thomson@otago.ac.nz

**Keywords:** dietary patterns, diet quality, obesity, dental caries, early childhood, toddlers, child (List three to ten pertinent keywords specific to the article, yet reasonably common within the subject discipline.)

## Abstract

We examined associations between dietary patterns at 12 months, characterised using multiple methodologies, and risk of obesity and early childhood caries (ECC) at 24–36 months. Participants were Australian toddlers (n = 1170) from the Study of Mothers’ and Infants’ Life Events affecting oral health (SMILE) birth cohort. Principal Components Analysis (PCA) and the Dietary Guideline Index for Children and Adolescents (DGI-CA) were applied to dietary intake data (1, 2 or 3-days) at 12 months, and regression analysis used to examine associations of dietary patterns with body mass index *Z*-score and presence of ECC at 24–36 months. Two dietary patterns were extracted using PCA: *family diet* and *cow’s milk*
*and*
*discretionary combination*. The mean DGI-CA score was 56 ± 13 (out of a possible 100). No statistically significant or clinically meaningful associations were found between dietary pattern or DGI-CA scores, and BMI *Z*-scores or ECC (n = 680). Higher *cow’s milk and discretionary combination* pattern scores were associated with higher energy and free sugars intakes, and higher *family diet* pattern scores and DGI-CA scores with lower free sugars intakes. The association between dietary patterns and intermediate outcomes of free sugars and energy intakes suggests that obesity and/or ECC may not yet have manifested, and thus longitudinal investigation beyond two years of age is warranted.

## 1. Introduction

Childhood obesity and early childhood caries (ECC) are two highly prevalent conditions affecting children [1,2]. In Australia, 26% of children have overweight or obese [3] and 34% have ECC by five years of age [4]. Given the commonality of both these conditions in early childhood, and evidence of an association between the two [5], the case has been made that obesity prevention could be targeted within the dental remit [6,7]. The Common Risk Factor Approach (CRFA) seeks to target risk factors common between conditions [8], with potential efficiencies and benefits over utilizing an approach that is disease-specific [8]. Thus, adopting a collaborative approach to tackling both childhood obesity and ECC may be more effective than an individual condition approach.

Poor diet in early life is a major modifiable risk factor for both obesity and ECC. This is particularly true for high intakes and frequencies of free sugars [9,10,11]. For example, an Australian study of over 4000 children aged 4–8 years found that obesity and ECC were both associated with sugary drink consumption [12]. Given that the first years of life are a critical period when dietary behaviours and food preferences are established, persisting and predicting disease in adulthood [13,14], understanding risk-promoting dietary factors and how they interact is important to inform interventions that aim to simultaneously prevent obesity and caries in early childhood.

Beyond sugar, poor diet comprises many other risk factors for obesity (energy, saturated fat, refined grains, processed meat) and ECC (e.g., acidic foods, protective role of dairy foods) [15,16]. Therefore, it is most appropriate to examine whole diet, rather than single nutrients (e.g., sugars) or foods (e.g., sugary drinks), as a means of understanding obesity and dental caries prevention in early life. Measuring multiple dietary components together acknowledges the synergistic and correlational nature of single foods and nutrients, and reflects how people eat [17,18]. Whole-of-diet patterns can be derived using an a priori score-based approach or an a posteriori data-driven approach. Score-based (a priori) approaches are commonly based on prevailing dietary guidelines which incorporate pre-existing knowledge of several chronic diseases, whereas data-driven approaches (a posteriori) use statistical data reduction techniques such as Principal Components Analysis (PCA) to explain the total variation in diet of a population [17,18]. Each approach has advantages and disadvantages over the other. For example, as a posteriori methods generate patterns based on empirical data, without *a-priori* knowledge, they do not necessarily represent optimal patterns [19], whereas a priori approaches do not account for biological interaction among nutrients [19]. Given this, examining diet using a combination of a priori and a posteriori approaches may provide greater insight into the diet-disease relationship. Doing so using the same sample can strengthen the understanding of this relationship for both obesity and ECC.

Whole-of-diet patterns have been characterised in young children using both a priori and a posteriori approaches and their association with later health outcomes investigated [20]. Studies exploring the relationship between dietary patterns and child obesity have shown inconsistent findings [20,21], while studies exploring the relationship between dietary patterns and ECC are rare. Those few studies which have investigated the latter relationship using a posteriori-derived indices have reported inconsistent findings [22,23] with only one study, to our knowledge, (and in an Asian multi-ethnic population), doing so using a priori-derived dietary patterns (exploratory factor analysis to examine diet trajectories between 6 and 12 months of age and ECC at ages 2 and 3 years) [24]. Such inconsistencies in the evidence may be due to discrepancies in the methodological approaches used to characterize dietary patterns (a priori vs. a posteriori). Thus, using multiple methodological approaches to characterize dietary patterns in early life would be advantageous in comparing and contrasting findings, and it would provide a more detailed dietary picture for that particular population.

Given the inconsistencies in findings and scarcity of research examining the relationship between whole-of-diet in the first years of life and the outcomes obesity and ECC, (along with the benefits of utilising multiple approaches to characterize diet and the greater recognition of the CRFA), this study aimed to investigate the association between whole diets and obesity and ECC in young children. The primary aim was to describe dietary patterns in Australian toddlers at 12 months of age using a priori and a posteriori approaches and then to examine their association with obesity and ECC at 24–36 months of age. Secondary aims were to examine associations between dietary patterns and free sugars and total energy intake, and to identify the socio-demographic determinants of observed dietary patterns.

## 2. Materials and Methods 

### 2.1. Data Source

This was a secondary analysis of data collected as a part of the Study of Mothers’ and Infants’ Life Events affecting health (SMILE), a prospective cohort study following socio-economically diverse South Australian newborns from birth to 24–36 months, described in detail elsewhere [25]. Briefly, mother–infant dyads were recruited from three major maternity hospitals between July 2013 and August 2014. Recruitment took place on postnatal wards by trained dental nurses and dental therapists within 48 h of birth. Mothers were provided with a written and verbal description of the study and only those mothers with sufficient English to comprehend the study instructions and who intended to remain living within the Adelaide region for at least 12 months were invited to participate. In total, 2147 mothers and 2181 infants, including 34 sets of twins, were recruited [25]. Children with complete dietary data (1, 2 or 3 days), socio-economic variables and outcome data were included in the secondary analysis. 

### 2.2. Data Collection and Entry

#### 2.2.1. Dietary Data

Children’s dietary intake was collected via 1 × 24 h dietary recall (24 HDR) and 1 × 2-day estimated food record over a non-consecutive 10-day period (week and weekend days) at 12 months of age [26]. Following their child’s first birthday and prior to data collection, mothers were mailed a food record and an accompanying letter advising them that a researcher would contact them to conduct the 24 HDR and to provide instructions on how to complete the food record [26,27]. A trained dietitian conducted the 24 HDR via telephone using a five-step multi-pass method [28]. Following the recall, dietitians allocated each mother two days in the subsequent week (10-day period) to record their child’s dietary intake in the food record. The food record contained a one-day food record example, and pictures of food portion sizes and common household measures to assist mothers in estimating serve sizes in the food recall and record [26,27].

Dietary data were double-entered into Foodworks 8 by a team of trained nutritionists/dietitians (Xyris Software, 2012–2016, Brisbane, QSL, Australia) as paired (3-day; 1 × 24 HDR and 1 × 2-day food record) and unpaired data (single 24 HDR) using data entry protocols for standardisation. Participants were included if they had at least one complete day of dietary intake data available. Data were analysed using the AUSNUT 2011–2013 food composition database [29], and food groups and servings were classified using the Australian Dietary Guideline (ADG) numeric classification system [25,26]. Owing to the rapidly changing infant and toddler food market, some infant foods (*n* = 187) consumed by the SMILE cohort did not exist in the AUSNUT 2011–2013 database. Accordingly, these foods were added to the database using information sourced from manufacturer’s websites and product nutrition information panels, mapped to a similar product in AUSNUT 2011–2013 for missing micronutrient values, and an appropriate 8-digit AUSNUT code allocated [26,27] (see Devenish et al. 2019 [30] for more details).

#### 2.2.2. Child and Maternal Sociodemographic Data

Sociodemographic data (including maternal age, smoking status, pre-pregnancy weight status, education attainment, annual household income and child gender) were collected from mothers at recruitment via standardised self-completed questionnaires [25]. Data on child age (at 24 HDR), breastfeeding duration and the age at which children were introduced to solids were obtained from a series of postal, online and face-to-face surveys at 3, 6, 12 and 24 months [25].

#### 2.2.3. Dental Data

Once children reached 24 months of age, they were invited (with their mothers), to participate in a dental examination. Dental examinations were conducted by a small team of specially trained dentists under standardised conditions [31,32]. Standard clinical indices were used; these had been developed at the Australian Research Centre for Population Oral Health (ARCPOH) and based on US National Institute of Dental and Craniofacial Research (NIDCR) protocol and the International Caries Detection and Assessment System (ICDAS) [33] were used. Non-cavitated or cavitated carious lesions, filling, missing tooth surfaces due to decay, non-carious developmental conditions and gingival conditions were recorded [25]. In the current analysis, ECC was characterised as present or absent, based on the presence of 1 or more cavitated surfaces at age 2 years.

#### 2.2.4. Anthropometric Data

Child weight and height data were collected at the dental examination using standardised methodology and equipment including digital scales and a portable stadiometer [25]. Pre-pregnancy maternal height and weight were self-reported in the baseline questionnaire. Body mass index (BMI) (kg/m^2^) was calculated for both the mother and child and weight status was categorised using World Health Organization (WHO) age-and gender-specific cut-offs [25].

### 2.3. Dietary Pattern Analysis

#### 2.3.1. Principal Components Analysis (PCA)

*A-posteriori* dietary patterns were extracted using Principal Components Analysis (PCA). Prior to this, a total of 2303 food and beverage items were categorised into meaningful and interpretable groups for use as input variables. Items were grouped based on their nutrient profile, the WHO free sugars recommendations [34], the Australian Guide to Healthy Eating (AGHE) [35] and the Australian Dietary Guidelines (ADG) [36]. Food groups that were not representative of usual toddler intake (i.e., dietary supplements, cooking agents and alcoholic beverages) were excluded. Fifty-eight food groups were created and used as input variables for PCA. 

PCA was conducted on all available dietary intake data. That is, participants with 1, 2 or 3-days of dietary data were included. Multiple days of dietary data were combined using mean intake across the days involved. The correlation matrix, Kaiser–Myer–Olkin measure of sampling adequacy, Bartlett’s test of sphericity and communalities were all inspected to determine whether a meaningful PCA could be performed [37]. All criteria were met demonstrating that all food groups contributed to the extracted patterns and, therefore, none were removed for subsequent analysis. Orthogonal (varimax) rotation was used to improve interpretability of the component loadings. The number of dietary patterns retained was determined using eigenvalues >1.5 and examination of a break point in the scree plot [37]. Based on the scree plot a two-factor (first break) and a seven-factor (second break) solution were considered. The seven factor solution did not meet the criteria for eigenvalues >1.5 and was not interpretable and thus not considered further. Patterns were named according to foods with loadings >0.25 to aid interpretability [38,39]. A dietary pattern score was calculated for each participant for each pattern by summing the product of a standardised gram of each item consumed by its factor loading. Patterns were normally distributed (mean 0, SD 1).

#### 2.3.2. The Dietary Guideline Index for Children and Adolescents (DGI-CA)

A priori dietary patterns were derived by applying the DGI-CA, a validated diet quality index score for children and adolescents aged 4–16 years [40], updated [41] to reflect adherence to the 2013 ADGs and extended here to include the recommended daily serves for 12–24-month-olds [42]. The tool comprises 11 indicators based on compliance with the dietary guidelines; one reflects diet variety, one reflects dietary moderation and nine reflect diet adequacy and diet quality. The total DGI-CA score is a sum of each indicator; it is converted to a score between 0 and 100, with higher scores reflecting greater adherence to the dietary guidelines. 

### 2.4. Statistical Analysis

Statistical analysis was conducted using SPSS version 25 (IBM SPSS Statistics for Windows, New York, NY, USA). Categorical data are presented as frequency (%) and continuous data as mean ± SD (as all were normally distributed). A Chi-square test (categorical) and independent t-test (continuous data) were employed to assess the statistical difference in sample characteristics between the total sample (n = 1170) and the final regression analysis sample (n = 680, see below). Standard linear regression was employed to investigate the associations between PCA-derived dietary patterns and DGI-CA scores with various outcome variables; (1) socio-demographic factors, (2) free sugars intake (grams), (3) energy intake (kilojoules) and (4) BMI *Z*-scores. Binary logistic regression was employed to investigate the relationships of dietary patterns and DGI-CA scores with the presence of dental caries. Multivariable models were adjusted for the following confounders: IRSAD (Index of Relative Socio-Economic Advantage and Disadvantage) deciles, one of four Socio-Economic Indexes for Areas (SEIFA) indices that ranks areas across Australia on a continuum of disadvantage (lowest score = 1) to advantage (highest score = 10), as a proxy for socio-economic position (SEP) [43], annual household income, maternal age, smoking status during pregnancy, weight status, education attainment, number of parents in the household, child’s age (at 24 HDR) and gender, and child’s age at which introduced to solids and at which breastfeeding ceased [38,40,44,45]. Due to missing data, the final regression analysis sample size was *n* = 680, which met the required sample size of *n* = 225 for assessing 15 predictors based on the ratio of 15 participants to 1 predictor [37]. Regression assumptions were checked through assessing the normality, linearity and variance of residuals [37]. The level of statistical significance was set at *p* < 0.05. To determine the magnitude of the associations, unstandardized regression coefficients (B), odds ratios and 95% confidence interval (CI) were used. 

## 3. Results

The characteristics of children and their mothers are described in Table 1. Dietary intake data were provided for 1170 children (46% female) with a mean age (at 24 HDR) of 13.1 (range 12–22) months. Most (62.1%) children were healthy weight, with 6.2% having overweight, 2.0% having obesity, and 0.4% underweight (29% missing). Few (8.8%) children had dental caries. The majority of mothers were partnered, university educated, employed prior to giving birth, not overweight or obese, and non-smokers during pregnancy. Mothers who provided complete dietary data but did not provide complete baseline or dental or weight outcome data were younger (analysis sample vs incomplete sample, mean = 30.8 vs. 30.1 years), and more educated (57.9% vs. 49.2% attended some university or above) but otherwise largely representative of the sample with complete data (Table 1). 

### 3.1. A Posteriori and A Priori Dietary Patterns

Two a posteriori dietary patterns were extracted using PCA (Table 2). The proportion of variance explained by both patterns was 7.9%. The first was characterised by vegetables, fresh fruit, non-white bread, cheese and non-discretionary red meat and poultry, all of which are common family-based items. Hence, this pattern was termed a *family diet* pattern. The second pattern was termed *cow’s milk and discretionary combination* because it was characterised by cow’s milk, fluoridated water, white bread, cheese and discretionary foods including processed meat, sugary products, sugar sweetened beverages and discretionary potato products. A priori dietary patterns were determined using the DGI-CA. The mean ± SD DGI-CA score was 56 ± 13 with 0% of children achieving the maximum possible score of 100 (data not shown). There were no statistically significant or meaningful differences between the final analysis sample and incomplete sample for dietary pattern scores (*family diet*: analysis sample vs incomplete sample, mean = 0.0 ± 0.9 vs. 0.0 ± 1.1; *cow’s milk and discretionary combination*: 0.0 ± 1.0 vs. 0.0 ± 1.0) and DGI-CA scores (57 ± 13 vs. 55 ± 13).

### 3.2. Association between Dietary Patterns and Adiposity or Dental Caries

After adjustment for covariates, no significant or clinically meaningful associations were found between dietary pattern scores or DGI-CA scores and BMI *Z*-scores or presence of dental caries (Table 3). 

### 3.3. Association between Dietary Patterns and Free Sugars or Energy Intakes

Dietary pattern scores and DGI-CA scores were independently associated with free sugars and energy intake at 12 months after adjusting for covariates (Table 4). A 1 SD higher *family diet* pattern score was associated with a 5.5 g lower free sugars intake, while a 10 point higher DGI-CA score would equate to a 4 g lower free sugars intake. Conversely a 1 SD higher *cow’s milk and discretionary combination* pattern score was associated with a 9.7 g higher free sugars intake, and a 1310 kJ greater energy intake, almost double that predicted by higher family diet pattern scores (779kJ). A 10 point higher DGI-CA scores was associated with a 380 kJ higher energy intake. 

### 3.4. Predictors of Dietary Patterns and Quality 

After adjusting for covariates, several maternal (age, education, employment status, IRSAD decile, household income) and child (age at 24 HDR, age of introduction to solids, breastfeeding duration) characteristics were independently associated with dietary pattern scores and/or DGI-CA scores (Table 5). For example, some maternal university education was associated with a 0.16 higher *family diet* pattern score than for those mothers whose highest education was the completion of school or vocational education. Higher scores on this *family diet* pattern were also associated with greater socioeconomic advantage and younger mothers. Conversely, higher scores on the *cow’s milk and discretionary combination* pattern were observed with older children at 24 HDR, earlier introduction to solids, earlier breastfeeding cessation, younger mothers and mothers who were unemployed prior to giving birth. Similarly, higher DGI-CA scores were observed with older children at 24 HDR, greater household income and university educated mothers. 

## 4. Discussion

The present study aimed to characterise whole-of-diet patterns at 12-months of age using multiple dietary pattern methods, and investigate their relationship with obesity and ECC at 24–36 months of age. No association was found between dietary patterns and obesity, or between dietary patterns and dental caries, using either the a priori DGI-CA, or a posteriori PCA approaches. However, higher free sugars and energy intakes, risk factors for both obesity and ECC, were positively associated with the *cows milk and discretionary combination* pattern (reflecting poorer-quality diet), and lower free sugars intake positively associated with both the *family diet* pattern and higher DGI-CA scores (both reflecting higher-quality diet). Dietary patterns were also predicted by maternal socio-demographic factors and child early feeding experiences in expected directions. Given this face validity of extracted dietary patterns, and their association with intermediate outcomes of free sugars and energy intake, the association between dietary patterns obesity and/or ECC may not yet have manifested and thus further longitudinal follow-up is warranted.

The lack of association between diet, obesity and dental caries in the present study is unsurprising given the inconsistent findings reported previously from investigations of both a priori and a posteriori-derived dietary patterns and measures of weight status [21,38,46,47,48] and dental caries [49,50,51,52] in early life. A possible explanation for the lack of association seen in the present study is that it may be too early to detect the influence of poor diet. For example, a 2014 systematic review found a positive association between energy-dense, high-fat and low-fibre dietary patterns in childhood and later overweight and obesity in high quality studies with a follow up period greater than 2 years (2–21 years) [53]. Thus, longer-term follow-up, beyond 24–36 months of age (i.e., one-year follow-up) is required. This is supported by the fact that dietary patterns were predicted by free sugars and energy intakes, as well as maternal factors (age, income, employment and education status) and early feeding experiences (breastfeeding cessation, age of introduction to solids) in expected directions, demonstrating face validity of the dietary patterns extracted using both methods. Therefore, further investigation of the relationship between dietary patterns, obesity and ECC, using the CRFA model, is warranted to clarify the utility of whole-of-diet approaches in examining obesity and dental caries in early life, or whether it is too soon in the developmental trajectory to identify any association.

The diets in this sample were similar to those observed previously in Australian [38], European [45], and US [44] toddlers. For example, the *family diet* pattern resembles the *14-month and 24-month core food* patterns in 14 (*n* = 552) and 24 (*n* = 493)-month-old Australian toddlers respectively, while the *cow’s milk and discretionary pattern* shares similarities with the *basic combination* (14 months) and *non-core pattern* (24 months) [38]. Furthermore, while the DGI-CA has not been previously applied to the diets of children under 4 years, the mean DGI-CA score (56/100, indicating poor adherence to the ADGs) is also consistent with previous research in older Australian children and adolescents (4–16 years, 54/100; 4–7 year olds 61/100 [40]; 4–8 years, 51/100; 4–16 years, 54/100; [41]). 

Despite no significant longitudinal association between diet and weight and dental status, both a priori and a posteriori-derived dietary patterns were associated cross-sectionally with free sugars and energy intakes; those are key risk factors for both obesity and ECC. That is, higher and lower free sugars intakes were associated with ‘poorer’ and ‘higher’ diet patterns, respectively. These findings are consistent with previous research in British [54] infants (*n* = 6065, 6 and 8 months of age) and Australian children and adolescents (*n* = 3416, 4–16 years) [40]). In contrast, although higher energy intakes were associated with poorer-quality dietary patterns (characterised a posteriori), they were also associated with higher-quality dietary patterns (characterised a priori [DGI-CA] and a-posteriori). These findings suggest that nutrient adequacy may be coupled with higher energy intakes, contradicting the currently understood concept that higher energy intakes and nutrient inadequacy occur simultaneously [40]. Further, given the association between poorer-quality dietary patterns and higher free sugars and energy intakes, it is possible that an association of dietary patterns with obesity and/or ECC could be observed with further longitudinal investigation.

Dietary patterns characterised in the present study were also associated with key socio-demographic characteristics. For example, poorer diets (derived a posteriori using PCA) were associated with earlier breastfeeding cessation and age of introduction to solids. This is consistent with previous findings in Australian [38] and European [44] toddlers and is likely explained by the influence of early feeding experiences (for example, flavour exposure from breastmilk) on later food taste and food acceptance [55,56]. Furthermore, earlier introduction of solids may coincide with an earlier introduction of discretionary foods [57], or an enhanced preference for discretionary-type foods [58]. Also consistent with previous research, poorer and higher-quality diets were associated with lower maternal employment status [40] and maternal income [40,44,59] and university-education [38,44,45,60], respectively, whilst both were associated with younger mothers [38]. Together, these findings support the well-established role of mothers in determining early experiences with health behaviours, including food and eating [21,61], highlighting potential intervention targets. Lastly, although previous research suggests diet quality decreases with child age [13,40], the present study found that older children tended to have poorer diet-quality patterns (derived a posteriori using PCA) but higher diet quality scores (derived a priori using the DGI-CA). However, the DGI-CA does not account for breast- or formula-milk and thus younger children may receive a lower overall DGI-CA score than older children who consume a greater amount and variety of foods other than milk.

### 4.1. Strengths and Limitations

This study is strengthened by the use of longitudinal data from the SMILE study which overcomes the possibility of reverse directionality between dietary patterns and obesity and ECC outcomes. The use of two dietary pattern approaches to assess relationships is novel and strengthens the findings, given that each provides different information about diet and the advantages and disadvantages of each approach are somewhat counteracted by the other. The use of a validated dietary index, the DGI-CA, and multivariable analysis, which has not been utilised in previous studies within the dental context, are also strengths. Lastly, despite a modest sample size, the most socially disadvantaged families were only slightly under-represented (15–17% of participants in the lowest two deciles). This is much better than most studies of this kind which are typically skewed towards the upper deciles. Nonetheless, the findings must be considered within the context of the study limitations. Energy intake was able to be estimated at only one time point (12-months of age) due to the dietary assessment method used (24-HDRand food records at 12 months), and thus only cross-sectional associations between dietary patterns and underlying free sugars and energy intakes could be investigated, precluding the determination of directionality. Another consideration is that some dental-specific (e.g., fluoride exposure, although it is unlikely that this would have varied greatly in the present sample as all were living in fluoridated areas at the time of recruitment) and obesity-specific (e.g., physical activity level) confounders were not able to be adjusted for in the multivariable analysis because they were not measured. Lastly, the limitations of PCA and the DGI-CA cannot be ignored. Principal Component Analysis is a highly subjective, multi-step process where numerous decisions are made by the researcher (for example, how data are grouped, how variables should be treated and how many patterns to retain), thus affecting the nature of the extracted patterns [18]. The DGI-CA is also limited in that it represents current dietary guidelines with disease outcomes considered individually, not concurrently. However, multiple dietary pattern approaches were utilised to account for the limitations of each.

### 4.2. Implications for Future Practice and Research

Although this study investigated the relationship between diet and obesity and ECC longitudinally, longer-term follow-up is warranted beyond two years of age to more fully determine the nature of the associations. While it was valuable to investigate the impact of dietary patterns characterised by PCA (a-posteriori) and the DGI-CA (a-priori) using the CRFA, it appears that exploration of dietary patterns using a more specific and tailored whole-of-diet approach might be more insightful. Reduced rank regression (RRR) and disease-specific indices are two methods that have potential in determining associations between whole diet and health outcomes such as obesity and dental caries in future studies [62]. Reduced rank regression has rarely been adopted in nutrition epidemiology; however, in the existing literature RRR-derived dietary patterns have been associated with obesity in children [48] and with cardiometobolic health in infants. Additionally, the associations between dietary patterns, free sugars and energy intakes identified in the present study highlight the potential for these nutrients to be used as response variables in RRR. Disease-specific indices offer another direction moving forward in determining the association of whole-of-diet in early life on later health outcomes, particularly in the dental context. However, to our knowledge, only one dental-specific index has been developed for the paediatric population [49], and this failed to account for the synergistic effects of positive (risk-reducing) nutrients such as calcium and phosphate (i.e., from dairy). Thus, a dental-specific index incorporating both negative (risk-promoting; free sugars) and positive (risk-reducing; calcium, phosphate) nutrients would be a useful tool to better understand the association between early life diet and ECC. Together, the use of RRR and disease-specific indices in future research could improve our understanding of the relationship between early life diet and childhood obesity and dental caries, and could ultimately inform new dietary guidelines for both obesity and dental caries prevention. Such guidelines could be useful for dentists, dietitians, child health nurses, public health workers and policymakers.

## 5. Conclusions

In summary, this is one of the first studies to describe early life diet using multiple methodologies and to investigate its impact on two key conditions prevalent in childhood, obesity and early childhood caries (ECC). While no association was found between a priori and a posteriori described dietary patterns, and obesity or ECC, the association with intermediate outcomes of free sugars and energy intakes is important, suggesting that obesity and/or ECC may not yet have manifested due to the short follow-up period (24–36 months). Thus, further longitudinal investigation beyond 2 years of age is warranted. Future studies utilising reduced rank regression and disease-specific indices to characterise whole diet may also contribute to a greater understanding of the impact of early life diet on obesity and ECC outcomes. This could ultimately assist in the design of new dietary guidelines and recommendations, useful for dentists, dietitians, child health nurses, public health workers and policymakers, and help to combat both childhood obesity and early childhood caries. 

## Figures and Tables

**Table 1 nutrients-11-02828-t001:** Characteristics of mother-child dyads in the sample with complete dietary data and sample with complete outcome and explanatory variables.

Characteristics	Total Sample ^a^		Analysis Sample ^b^	
	*n* = 1170, n (%)		*n* = 680, *n* (%)	
Child Characteristics				
Age at time of 24 HDR (months) ^c,d^	13.1	0.9	13.1	0.8
Gender				
Female	540	46.2	313	46.0
Breastfeeding duration (weeks) ^e^				
<17	254	21.7	176	25.9
17–25	107	9.1	80	11.8
26–51	261	22.3	198	29.1
>52	316	27.0	226	33.2
Age of introduction to solids (weeks) ^f^				
<17	235	20.1	147	21.6
17–25	692	59.1	454	66.8
>26	121	10.3	79	11.6
Weight status ^g,h^				
Underweight	5	0.4	5	0.7
Healthy weight	727	62.1	593	87.2
Overweight	73	6.2	59	8.7
Obesitye	23	2.0	23	3.4
Maternal and household characteristics				
Age at birth ^e,f^	30.5	5.1	30.8	5.0
Household income ^c,d^ (annual)				
<40,000	162	14.0	83	12.2
40,100–80,000	369	31.8	216	31.8
80,100–120,000	357	30.8	216	31.8
>120,000	272	23.4	165	24.3
IRSAD ^d,i^				
Deciles 1–2 (most disadvantaged)	195	16.7	98	14.4
Deciles 3–4	247	21.1	150	22.1
Deciles 5–6	235	20.1	137	20.1
Deciles 7–8	220	18.8	141	20.7
Deciles 9–10 (most advantaged)	262	22.4	154	22.6
Two parent household ^f^				
Yes	1090	93.2	641	94.3
Smoking status during pregnancy ^d^				
Yes	78	6.7	37	5.4
Weight status ^d,i^				
Underweight	47	4.0	28	4.1
Healthy	572	48.9	350	51.5
Overweight	262	22.4	168	24.7
Obesitye	211	18.0	134	19.7
Education attainment ^d^				
High school/vocational	524	44.8	286	42.1
Some university or above	635	54.3	394	57.9
Work status prior to birth ^d^				
Employed	875	74.8	531	78.1

^a^ Sample with complete dietary data but incomplete outcome or explanatory variables. ^b^ Sample with complete outcome and explanatory variables. ^c^ Values presented as mean (SD). All other values presented as frequency (%). ^d^ Less than 5% missing data. ^e^ 20% missing data. ^f^ Less than 15% missing data. ^g^ 29% missing data. ^h^ Weight status categories according to body mass index (BMI) (kg/m^2^); Underweight <18.5 kg/m^2^, Healthy weight 18.5–24.9 kg/m^2^, Overweight 25–29.9 kg/m^2^, Obese >30 kg/m^2^. ^i^ IRSAD: Index of Relative Socio-Economic Advantage and Disadvantage. HDR: hour dietary recall.

**Table 2 nutrients-11-02828-t002:** Varimax-rotated food group loadings on each of the dietary patterns at 12 months extracted by Principal Components Analysis (PCA).

Foods	Percent Consumingd (%) ^b^	Dietary Patterns ^a^	
		Family Foods	Cow’s Milk Discretionary
Vegetables: other non-discretionary	64.4	**0.534**	0.086
Fruit: Fresh	90.0	**0.513**	0.073
Vegetables: Orange	60.2	**0.502**	−0.154
Vegetables: Green and Brassica	56.2	**0.488**	−0.034
Meat: Red	28.9	**0.305**	0.005
Poultry and feathered game	39.3	**0.295**	0.006
Bread: non-white	42.1	**0.290**	0.247
Potato: non-discretionary	42.2	**0.260**	−0.015
Cereal: Breakfast with no added sugar	58.9	0.250	−0.045
Infant food: commercial sweet	62.0	−0.238	0.043
Flours and Grains	60.4	0.225	−0.207
Fish: non-discretionary	21.2	0.223	−0.076
Infant food: commercial savoury	13.3	−0.221	−0.152
Dairy yoghurt: whole fat	19.6	0.219	−0.040
Eggs	28.3	0.216	0.002
Sauces and condiments	28.6	0.201	0.097
Fruit juice: non-discretionary	9.7	−0.186	0.074
Fruit juice: discretionary	3.7	−0.142	0.045
Infant drinks	1.8	−0.137	−0.007
Dairy milk: reduced fat	3.8	−0.130	0.130
Beverages: other discretionary	1.8	−0.091	0.001
Vegetables MD: discretionary	3.0	0.053	0.034
Fruit: stewed and MD	3.2	0.051	0.035
Fruit: packaged with added sugar	1.8	0.049	0.015
Meat: MD non-discretionary	20.6	0.033	−0.028
Vegetables: MD non-discretionary	3.8	−0.032	−0.030
Dairy milk: flavoured	1.9	0.029	0.002
Infant formula and breastmilk	71.0	−0.207	**−0.682**
Dairy milk: Whole fat	71.0	−0.007	**0.504**
Water: domestic	96.8	0.247	**0.426**
Meat: processed	32.8	0.010	**0.397**
Sugar and sugary products	23.4	−0.135	**0.330**
Cheese	57.5	**0.279**	**0.309**
Potato: discretionary	16.0	−0.188	**0.289**
Bread: white	48.7	−0.236	**0.272**
Sugar sweetened beverages: discretionary	1.5	−0.165	**0.259**
Margarine and oils	27.0	0.117	0.250
Sweet biscuits and cakes	40.9	−0.001	0.244
Fish: discretionary	6.8	0.036	0.234
Confectionary	6.7	−0.178	0.217
Nuts and seeds	19.7	0.147	0.211
Bread: sweet	9.5	−0.064	0.210
Fruit: dried	19.6	0.031	0.206
Cereal: other discretionary	35.1	−0.107	0.205
Cereal: savoury non-discretionary	43.0	0.039	0.173
Butter	38.6	−0.021	0.168
Water: other	2.4	−0.027	0.167
Fruit: packaged with no added sugar	8.3	−0.049	0.143
Flavoured yoghurt and custard	27.9	0.051	0.138
Poultry: MD non-discretionary	15.6	0.064	0.132
Cereal fruit and nut bars	4.7	−0.065	0.129
Dairy: other discretionary	9.6	−0.099	0.105
Poultry: MD discretionary	1.1	−0.046	0.096
Dairy Alternatives: non-discretionary	3.9	0.078	0.095
Soups	9.3	0.005	−0.091
Legumes and Beans	10.9	0.025	0.073
Cereal: Breakfast with added sugar	5.6	0.003	0.039
Meat: MD discretionary	0.3	−0.021	0.030

Abbreviations: PCA: principal component analysis; MD: mixed dishes. ^a^ Loadings ≥ 0.25 in bold to aid labelling of dietary patterns. ^b^ Total number of participants consuming the food or beverage.

**Table 3 nutrients-11-02828-t003:** Associations between dietary patterns at 12 months and BMI *Z*-scores, BMI-weight category, dental caries at 24–36 months, after adjusting for covariates ^a^.

	BMI *Z*-Score ^b^			Dental Caries ^c^			
	B	95% CI	*p*-Value	OR	Wald	95% CI	*p*-Value
Diet Patterns ^d^							
Family Diet	0.031	−0.059, 0.120	0.500	0.830	1.346	0.605, 1.137	0.246
Cow’s milk and discretionary combination	0.022	−0.068, 0.111	0.632	0.889	0.554	0.653, 1.212	0.457
DGI-CA scores *(Scale: 0–100)*	0.007	0.000, 0.013	0.053	0.987	1.287	0.966, 1.009	0.257

Abbreviations: BMI, body mass index, 95%CI, 95% confidence interval. ^a^ Maternal (age at birth, Index of Relative Socio-Economic Advantage and Disadvantage (IRSAD), household income, two parent household, smoking status, weight status, education attainment and work status prior to birth) and child (age at 24 HDR, gender, breastfeeding duration and age introduced to solids) confounders were adjusted for in analysis. ^b^ Results were obtained using standard linear regression models with child BMI *Z*-scores as outcome variable and diet pattern scores, DGI-CA scores and all respective covariates as independent predictors. ^c^ Defined as the presence of 1 or more cavitates surfaces. Results were obtained by binary logistic regression models with dental status as the outcome variable and diet pattern scores, DGI-CA scores and all respective covariates as independent predictors. ^d^ Dietary pattern scores are standardized scores with a mean and standard deviation of 0 ± 1.

**Table 4 nutrients-11-02828-t004:** Associations between dietary patterns at 12 months and intake of free sugars (g) and total energy intake (kJ), after adjusting for covariates ^a^.

	Free Sugars (g)			Energy Intake (kJ)		
	B	95% CI	*p*-Value	b	95% CI	*p*-Value
Diet Patterns ^b^						
Family Diet	−5.461	−7.088, −3.834	<0.001	779.112	461.090, 1097.134	<0.001
Cow’s milk and discretionary combination	9.767	8.140, 11.394	<0.001	1310.417	992.307, 1628.526	<0.001
DGI-CA scores (Scale: 0–100)	−0.393	−0.529, −0.258	<0.001	37.784	12.301, 63.266	0.004

Abbreviations: BMI, Body Mass Index, 95%CI, 95 Percent Confidence Interval. ^a^ Results were obtained using standard linear regression models with child BMI Z-scores as outcome variable and diet pattern scores, DGI-CA scores and all respective covariates as independent predictors. Maternal (age at birth, Index of Relative Socio-Economic Advantage and Disadvantage (IRSAD), household income, two parent household, smoking status, weight status, education attainment and work status prior to birth) and child (age at 24 HDR, gender, breastfeeding duration and age introduced to solids) confounders were adjusted for in analysis. ^b^ Dietary pattern scores are standardized scores with a mean and standard deviation of 0 ± 1.

**Table 5 nutrients-11-02828-t005:** Multivariable associations with dietary patterns at 12 months and mother-child dyad characteristics, after adjusting for covariates (*n* = 680) ^a^.

	Dietary Patterns ^i^						DGI−CA ^i^		
	Family Diet			Cow’s Milk and Discretionary Combination					
	B	95% CI	*p*-Value ^j^	B	95% CI	*p*-Value ^j^	b	95% ci	*p*-Value ^j^
Child Characteristic									
Gender	−0.101	−0.242, 0.040	0.160	0.023	−0.117, 0.164	0.745	−1.079	−2.938, −0.044	0.255
Age ^b^	0.067	−0.017, 0.151	0.116	0.179	0.095, 0.263	0.000	1.661	0.553, 2.769	0.003
Breastfeeding duration	0.036	−0.029, 0.100	0.275	−0.098	−0.162, −0.033	0.003	0.560	−0.291, 1.410	0.197
Age of introduction to solids	0.103	−0.026, 0.233	0.118	−0.175	−0.305, −0.046	0.008	−0.107	−1.816, 1.601	0.902
Maternal Characteristics									
Age at child’s birth	−0.019	−0.033, −0.004	0.012	−0.024	−0.039, −0.010	0.001	−0.130	−0.324, 0.064	0.189
IRSAD Decile ^c^	0.033	0.006, 0.060	0.016	0.002	−0.025, 0.029	0.867	0.221	−0.136, 0.577	0.225
Household Income ^d^	0.071	−0.011, 0.152	0.090	0.078	−0.004, 0.159	0.062	1.967	0.888, 3.046	0.000
Two parent household ^e^	0.188	−0.128, 0.505	0.243	−0.155	−0.471, 0.161	0.336	0.304	−3.873, 4.480	0.887
Smoking status during pregnancy ^e^	−0.259	−0.577, 0.058	0.109	−0.039	−0.356, 0.278	0.811	−3.743	−7.931, 0.445	0.080
Weight status (Healthy weight) ^f^									
Underweight	0.055	−0.305, 0.416	0.764	−0.304	−0.665, 0.056	0.098	2.277	−2.482, 7.035	0.348
Overweight	−0.048	−0.223, 0.127	0.591	0.007	−0.665, 0.056	0.934	−0.480	−2.787, 7.035	0.683
Obese	−0.064	−0.255, 0.128	0.514	0.160	−0.032, 0.351	0.102	−1.168	−3.692, 1.827	0.364
Work status prior to birth ^g^	−0.095	−0.273, 0.082	0.290	0.192	0.015, 0.369	0.033	−1.434	−3.772, 0.905	0.229
Education attainment ^h^	0.160	0.005, 0.315	0.043	−0.146	−0.301, 0.009	0.064	2.594	0.548, 4.640	0.013

^a^ Results were obtained using standard linear regression models using diet pattern and DGI-CA scores as dependent variables and all respective covariates as independent variables. ^b^ Age at time of 24 HDR (12 months). ^c^ Index of Relative Socio-Economic Advantage and Disadvantage, IRSAD decile categorized as 1 = most disadvantaged and 10 = most advantaged. ^d^ Household income categorized as (1) <40 K (2) 40.1–80 K (3) 80.1–120 K (4) >120 K. ^e^ Categorised as yes (reference category) or no. ^f^ Weight status categories according to Body Mass Index (BMI) (kg/m^2^); Underweight <18.5 kg/m^2^, Healthy weight 18.5–24.9 kg/m^2^, Overweight 25–29.9 kg/m^2^, Obesity >30 kg/m^2^. Healthy weight is used as reference category. ^g^ Work status categorized as employed (reference category) or unemployed. ^h^ Education attainment categorized as school/vocational or some university and above (reference category). ^i^ Dietary pattern scores are standardized scores with a mean and standard deviation of 0 ± 1. DGI-CA scores are on a scale between 0–100. ^j^ Model Fit: All linear regression models statistically significant (*p* < 0.001). Bold values represent statistically significant values (*p* < 0.05).

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
