# Peer review of "Dietary Patterns and Risk of Obesity and Early Childhood Caries in Australian Toddlers: Findings from an Australian Cohort Study"

_nutrients, 2019, doi:10.3390/nu11112828_

Round 1
Reviewer 1 Report
The authors have analyzed dietary patterns of a children cohort at 12 months in relation to obesity and caries at 24-36. The study is well powered and could potentially provide interesting information. They found no association between the two dietary patterns identified by PCA or DGI-CA score and either disease. However, they found significant associations with energy and sugar intake. The authors conclude that these data suggest that the follow-up is not long enough for obesity and ECC to manifest and be detected in the analysis. This is indeed the main limitation of the manuscript (acknowledged by the authors). While the lack of association between the factors analyzed could be of interest, the data presented on the manuscript does is insufficient to make a conclusion on whether dietary patterns are associated or not with obesity and caries (only up to 2 years of age, which is too early). Data from 4-6 year-old children would allow the authors to make a conclusion on whether these dietary patterns are associated with obesity and caries or not.
Other major comments.
The PCA analysis and rotation is a nice tool used in this study. I believe showing the data for the PCA score plot, loading plots and scree plot would be a big help for interpreting the data. Also, the authors selected only 2 factors (based on eigenvalues > 1.5 and break point in the scree plot), but these factors accounted for only 7.9% of the variance. These is quite a low %, and leaves a big part of the variance unexplained. Are the other factors characterized by any dietary pattern even if with less foods and probably not conforming a “whole-of-diet” pattern? Could these other factors be associated with obesity or caries?
The authors analyzed BMI-SDS at 24 months as a measure of the obesity state. Given that crossing percentiles in BMI has been associated to risk of obesity, another interesting variable to look at could be the change in BMI-SDS between 12 and 24 months of age. Have the authors performed these associations?
Minor comments
It is not completely clear to this reviewer the number of subjects that were analyzed in the main objective of the study (Table 3). While I understand that it is n=680 (subjects with all data), the number stated in the abstract is n=1170 (total subjects). Could the authors clarify this?
The authors have done a good job adjusting the statistical models for different covariates. However, birth weight could be an important confounding factor in these analyses (especially related to obesity risk) and should be added to the models.
Reviewer 2 Report
The aim of the study was to examine associations between dietary patterns at 12 months of age and risk of obesity and early childhood caries at 24-36 months. This is an interesting, well-designed paper, and the results are clearly presented.
Comments:
Line 190 – please explain the abbreviation: IRSAD In the characteristics of the studied children, the authors should indicate what percentage were overweight and obese, and what underweight. It should also be stated what percentage of children was found to have caries. Did BMI of children with caries differ from BMI of children without caries? The authors should also explain whether there were children in the study group with diseases that required diet modifications, e.g. food allergies, and discuss how this could affect the results of the study.Author Response
Please see attachment

Round 2
Reviewer 1 Report
The authors clarified or answered most of the questions.
Minor comment: “scree plot” has been misspelled as “screen plot” in several places.